# Extent of N-Terminus Folding of Semenogelin 1 Cleavage Product Determines Tendency to Amyloid Formation

**DOI:** 10.3390/ijms24108949

**Published:** 2023-05-18

**Authors:** Daria A. Osetrina, Aleksandra M. Kusova, Aydar G. Bikmullin, Evelina A. Klochkova, Aydar R. Yulmetov, Evgenia A. Semenova, Timur A. Mukhametzyanov, Konstantin S. Usachev, Vladimir V. Klochkov, Dmitriy S. Blokhin

**Affiliations:** 1NMR Laboratory, Medical Physics Department, Institute of Physics, Kazan Federal University, Kremlevskaya Str., 18, Kazan 420008, Russiadmitr.blokhin@gmail.com (D.S.B.); 2Kazan Institute of Biochemistry and Biophysics, FRC Kazan Scientific Center, Russian Academy of Sciences, Kazan 420111, Russia; 3Laboratory of Structural Biology, Institute of Fundamental Medicine and Biology, Kazan Federal University, Kazan 420021, Russia; 4Laboratory for Structural Analysis of Biomacromolecules, Federal Research Center “Kazan Scientific Center of Russian Academy of Sciences”, Kazan 420111, Russia

**Keywords:** SEM1(49–107), SEM1(45–107), Semenogelin 1, amyloid, HIV, NMR spectroscopy, CD spectroscopy, DLS spectroscopy, ThT fluorescence, spatial structure

## Abstract

It is known that four peptide fragments of predominant protein in human semen Semenogelin 1 (SEM1) (SEM1(86–107), SEM1(68–107), SEM1(49–107) and SEM1(45–107)) are involved in fertilization and amyloid formation processes. In this work, the structure and dynamic behavior of SEM1(45–107) and SEM1(49–107) peptides and their N-domains were described. According to ThT fluorescence spectroscopy data, it was shown that the amyloid formation of SEM1(45–107) starts immediately after purification, which is not observed for SEM1(49–107). Seeing that the peptide amino acid sequence of SEM1(45–107) differs from SEM1(49–107) only by the presence of four additional amino acid residues in the N domain, these domains of both peptides were obtained via solid-phase synthesis and the difference in their dynamics and structure was investigated. SEM1(45–67) and SEM1(49–67) showed no principal difference in dynamic behavior in water solution. Furthermore, we obtained mostly disordered structures of SEM1(45–67) and SEM1(49–67). However, SEM1(45–67) contains a helix (E58-K60) and helix-like (S49-Q51) fragments. These helical fragments may rearrange into β-strands during amyloid formation process. Thus, the difference in full-length peptides’ (SEM1(45–107) and SEM1(49–107)) amyloid-forming behavior may be explained by the presence of a structured helix at the SEM1(45–107) N-terminus, which contributes to an increased rate of amyloid formation.

## 1. Introduction

Human immunodeficiency virus (HIV) was first identified in 1981 [1]. This is one of the most dangerous diseases due to its influence on cells of the human immune system. Over time, HIV causes acquired immunodeficiency syndrome (AIDS) which may lead to susceptibility to various infections and tumors and eventually death [2]. Since the main route of transmission of the virus is unprotected intercourse, seminal fluid is considered as the main factor in increasing HIV activity [3].

Human semen forms a coagulum immediately after ejaculation. Semenogelin 1 (SEM1) and Semenogelin 2 (SEM2) are predominant proteins in seminal vesicles occupying approximately 60% of the ejaculate volume [4,5,6,7,8]. Both proteins originate from the glandular epithelium of the seminal vesicles, which secrete them in high concentrations (60 g/L) [9]. Semenogelins 1 and 2 are the predominant structural proteins of the loose gel formed in freshly ejaculated human semen. The concentration of SEM1 is five to ten times higher compared with SEM2 in semen. Semenogelins perform many biochemical functions, such as semen liquefaction, antibacterial activity, protection and other functions of semen [9]. The Semenogelins participate in noncovalently linked gel-like structure formation that ensnares the spermatozoa. Within the first 20 min, the gel-like structure is liquefied by serine proteases, primarily prostate-specific antigen (PSA), which cleaves the full-sized proteins into small fragments [10].

In previous works, it was demonstrated that certain cationic polypeptides within the semen and seminal plasma could form amyloid fibrils amplifying HIV infection [11]. Semenogelin cleavage products found in semen in high concentration could be responsible for an increase in the activity of HIV virions in seminal fluid. The analysis of semen amyloid compounds showed the presence of SEM1 fragments (SEM1(45–107), SEM1(49–107), SEM1(68–107) and SEM1(86–107)) (Figure 1) [12,13].

It is believed that semen amyloid fibrils enhance the adhesion of HIV virions to cell membranes by decreasing electrostatic repulsion between membranes of the virus and target cell [14,15]. Additionally, studies have shown that amyloid fibrils from Semenogelin 1 play a physiological role in the process of fertilization [16]. The most obvious similarity between these processes is their same physiological circumstances [17]. The fusion of semen and oocyte membranes, as well as HIV and target cell membranes, is an energetically unfavorable process that requires many different cooperative protein–protein interactions [18,19,20].

Herein, we report the discovery of amyloidogenic peptides SEM1(45–107) and SEM1(49–107), which are derived from human SEM1 (Figure 1). A clear difference was observed in their amyloid-forming process rate via Thioflavin T (ThT) fluorescence spectroscopy. We observed a more intensive aggregation process of SEM1(45–107) in comparison with SEM1(49–107) via ThT fluorescence and TEM data. Interestingly, the primary structure of SEM1(45–107) differs from SEM1(49–107) only by the presence of four amino acid residues (G45Q46H47Y48) at N-domains. In this regard, the N-domains of SEM1(45–107) and SEM1(49–107) (SEM1(45–67) and SEM1(49–67), respectively) have been characterized using structural and dynamics NMR methods, circular dichroism (CD) and dynamic light scattering (DLS) spectroscopies.

Nuclear magnetic resonance spectroscopy (NMR) is an informative method for determining the spatial structure of proteins and peptides in solution [21,22,23,24,25,26]. Pulsed-field gradient NMR spectroscopy (PFG-NMR) and dynamic light scattering spectroscopy (DLS) are effective methods for studying the translational mobility of a protein in solution [27,28,29,30,31,32]. CD was applied for the verification of the protein secondary structure [33]. In our discussion, we will describe the difference in the amyloid behavior of SEM1 cleavage products based on an individual study of the N-domains (SEM1(49–67) and SEM1(45–67)) of SEM1(49–107) and SEM1(45–107). Based on structural studies of N-domains and the Molecular dynamics (MD) simulation of full-sized peptides, we suppose the responsibility of helical fragments in N-terminuses of SEM1(45–107) in fast amyloid formation. In a previous article, Lui et al. (2010) [34] reported monomer helixes’ rearrangement into amyloid β-strand structures during the formation of amyloid fibrils. Thus, the presence of helix-like structures of SEM1(45–107) may lead to an increase in the amyloid-forming rate in comparison with SEM1(49–107).

## 2. Results

### 2.1. Thioflavin T Fluorescence Measurements of SEM1(49–107) and SEM1(45–107)

Figure 2A depicts the ThT fluorescence spectra of SEM1(45–107) and SEM1(49–107) peptides and a control pure ThT solution (black line). The red and green lines show the ThT fluorescence spectra of the SEM1(49–107) and SEM1(45–107) solutions, respectively. The red and black lines are almost identical, pointing to the absence of the amyloid fibril formation of SEM1(49–107) in water solution. In the case of SEM1(45–107), ThT fluorescence intensity (green line) increases by 4–5 times, indicating the presence of amyloid fibrils in the peptide solution.

As mentioned above, only fresh peptide solutions (up to 10 min after synthesis) were used for ThT fluorescence measurements. Therefore, the fibril formation of SEM1(45–107) starts immediately after purification in comparison with SEM1(49–107). However, the increased ThT fluorescence intensity measured 30 min after purification showed the amyloid formation of both SEM1(49–107) and SEM1(45–107) peptides (see Appendix A). The time dependence of the fluorescence intensity of ThT showed less time for the amyloid formation of SEM1(45–107) in comparison with SEM1(49–107) (Figure 2B). We used transmission electron microscopy (TEM) data to detail the fibril formation process. TEM images of SEM1(45–107) and SEM1(49–107) fibrils were recorded an hour after preparation (Figure 3). TEM images show that the length of SEM1(45–107) fibrils is larger than that of SEM1(49–107) ones. Thus, TEM data are in good agreement with ThT data, where we observed the more intensive fibril formation of SEM1(45–107) compared with SEM1(49–107).

### 2.2. Nuclear Magnetic Resonance Spectroscopy

According to ThT fluorescence spectroscopy data, an unstable monomeric state of SEM1(49–107) and SEM1(45–107) peptides in water solution was shown, which made it impossible to study their structure via NMR spectroscopy. Seeing that the peptide amino acid sequence of SEM1(45–107) differs from SEM1(49–107) only by the presence of four additional amino acid residues in the N-domain, these domains of both peptides were obtained via solid-phase synthesis and studied via NMR spectroscopy. The ^1^H, ^15^N and ^13^C chemical shifts of SEM1(45–67) (BMRB-51673) and SEM1(49–67) (BMRB-51691) assignment were determined with the help of NMR spectra (2D ^1^H-^1^H TOCSY, ^1^H-^1^H NOESY, ^1^H-^13^C HSQC, ^1^H-^13^C HMBC, ^1^H-^13^C HSQC-TOCSY and ^1^H-^15^N HSQC (Figure 4 and Appendix A)). The experimental restraints for the structural calculation were determined from NMR data (Table 1 for SEM1(45–67); Table 2 for SEM1(49–67)). The ^3^J_Hα-HN_ scalar coupling constants were determined via the 2D JRES spectrum analysis. The values of the ^3^J_Hα-HN_ scalar coupling constant were used to determine ψ dihedral angle restraints via Karplus curves [35]. The H, N, H_α_, C_α_ and C chemical shifts were applied for the determination of dihedral angles using the TALOS+ service (https://spin.niddk.nih.gov/bax/nmrserver/talos, accessed on 25 November 2022). Distance constraints (Appendix A for SEM1(45–67), Appendix A for SEM1(49–67)) were determined with the help of ^1^H-^1^H NOESY spectra analysis, where high-intensity signals corresponded to short distances between nucleuses, and low-intensity signals corresponded to long distances.

The NOE data of SEM1(45–67) provide 143 distance restraints, including 88 intra-residual distances and 55 inter-residual distances (24 distances between sequential amino acid residues (i–j = 1), 16 distances between medium-distance amino acid residues (i–j = 2–6) and 15 distances between long-range ones (i–j > 6)) (Appendix A). The NOE data of SEM1(49–67) provide 67 distance restraints including 54 distances within amino acid residues and 13 distances between spatially close amino acid residues (8 distances between sequential amino acid residues (i–j = 1), 2 distances between medium-distance amino acid residues (i–j = 3) and 3 distances between long-range ones (i–j > 4)) (Appendix A).

The distances constraints, dihedral angles and ^3^J_Hα-HN_ scalar coupling constants were used as input parameters for the spatial structure calculation with the help of XPLOR-NIH (v. 3.6) software. Figure 5 shows structural ensembles of SEM1(45–67) (PDB ID: 8BOO) (Figure 5A) and SEM1(49–67) (PDB ID: 8BVZ) (Figure 5B). Ramachandran plots of SEM1(45–67) (Appendix A) and SEM1(49–67) (Appendix A) validated the obtained peptides’ structures [36]. Both peptides have mostly unordered structures and have a horseshoe shape, which is consistent with a large number of long-range distances (i–j > 6) of 15 and 3 for SEM1(45–67) and SEM1(49–67), respectively. DSSP analysis (http://bioinformatica.isa.cnr.it/SUSAN/NAR2/dsspweb.html, accessed on 11 January 2023) [37] of the calculated structures (Appendix A) showed that SEM1(45–67) contains 3_10_-helix (E58-K60) and ordered helix-like fragments (S49-Q51), while any ordered regions in the case of SEM1(49–67) were not observed (Appendix A). The presence of 3_10_-helix (E58-K60) was confirmed via the NOEs characteristics [38] (d_α,N_(i, i + 3): E58H_α_-G61H_N_, d_α,β_(i, i + 3): T57H_α_-K60H_β_).

### 2.3. Circular Dichroism Spectroscopy

Circular dichroism (CD) spectroscopy was used for the validation of the calculated peptides’ structures. The CD spectra of SEM1(45–67) (Figure 6A) and SEM1(49–67) (Figure 7A) are typical for the prevailing of an unordered polypeptide chain. The analysis of the CD spectra (Figure 6B and Figure 7B) using the DichroWeb web service provided the following secondary structure components: SEM1(45–67) consists of distorted helixes (24%), regular β-strands (2%), distorted β-strands (11%), turns (33%) and unordered structural fragments (30%); SEM1(49–67) consists of distorted helixes (6%), regular β-strands (3%), distorted β-strands (14%), turns (37%) and unordered structural (40%) fragments. The RMSD values of the CD spectra analysis were 0.008 and 0.017 for SEM1(45–67) and SEM1(49–67), respectively. Thereby CD data also confirmed the mainly disordered structure of both peptides obtained via NMR spectroscopy. The components of regular β-strands and distorted β-strands are in agreement with the horseshoe shape of the peptides obtained via the NMR structures.

### 2.4. PFG-NMR Spectroscopy of SEM1(45–67) and SEM1(49–67)

DOSY spectra of peptides SEM1(45–67) and SEM1(49–67) were recorded to confirm that the peptides were in monomeric forms. For SEM1(45–67) and SEM1(49–67), the mono exponential diffusion decays of signals at 8.02–7.94 ppm were observed (Figure 8). Diffusion decays were characterized by self-diffusion coefficients D (2.07 ± 0.02·10^−10^ m^2^/s SEM1(45–67)) and 2.35 ± 0.03·10^−10^ m^2^/s (SEM1(49–67)). We calculated the hydrodynamic radius *R_h_* using the Stokes–Einstein relation:(1)$$R_{h}=\frac{k_{B}T}{6\pi \eta D},$$
where *T* is the temperature, *k_B_* is the Boltzmann constant and *η* is the water viscosity. The calculated hydrodynamic radiuses are 1.2 nm and 1 nm for SEM1(45–67) and SEM1(49–67), respectively. These values correspond to monomer forms of peptides and are consistent with NMR structures of SEM1(45–67) and SEM1(49–67).

### 2.5. DLS Spectroscopy of SEM1(45–67) and SEM1(49–67)

The oligomerization of SEM1(45–67) and SEM1(49–67) was further monitored via dynamic light scattering (DLS), which enables the observation of NMR-invisible diffusive species since the scattering intensity is very dependent on the particle mass/size [39]. The evolution of the size distribution with time is shown in Figure 9.

The hydrodynamic radii *R* of SEM1(49–67) and SEM1(45–67) were obtained using the DLS technique. In Figure 9, one can see that SEM1(49–67) and SEM1(45–67) solutions are characterized by single peaks. Meanwhile, the increased peak width of SEM1(49–67) indicates a continuous size distribution of different conformational states. We estimated the hydro-dynamic radius of *R* = 1.8 nm and *R* = 1.2 nm for SEM1(49–67) and SEM1(45–67), respectively. Thus, the *R* values of peptides are in acceptable agreement with the calculated size of both peptides via the DOSY experiments and their spatial structures.

The hydrodynamic radius *R* values obtained via the DLS technique are usually strongly influenced by weak intermolecular interactions [40]. The major role of protein–protein interactions is caused by electrostatic interactions. Therefore, the *ζ* -potential values of the peptide solutions were determined. We did not find principal differences in the *ζ* -potential values of SEM1(49–67) (*ζ* = +11.84 mV) and SEM1(45–67) (*ζ* = +17.44 mV) solutions. On the basis of *ζ*-potential values, we calculated the peptide effective charges of SEM1(49–67) (Q = +2.33*e*) and SEM1(45–67) (Q = +2.28*e*), which were found to be close to the calculated ones (https://www.protpi.ch/Calculator/ProteinTool).

### 2.6. MD Simulation of SEM1(45–107) and SEM1(49–107)

All-atom simulations were performed for the evaluation of SEM1(45–107) and SEM1(49–107) spatial structures. The simulation process consisted of several stages. In the first stage, we created files containing experimental spatial constraints (internuclear distances and dihedral angles) collected from restraints of individual peptide fragments (SEM1(45–67) and SEM1(68–107) for SEM1(45–107) and SEM1(49–67) and SEM1(68–107) for SEM1(49–107)). Restraints for SEM1(68–107) were taken from previous work [21]. In the next step of the calculation, SEM1(45–107) and SEM1(49–107) structures were annealed in the XPLOR-NIH (v. 3.6) program, where experimental spatial constraints were used as input parameters. SEM1(45–107) and SEM1(49–107) structures were calculated according to the protocol described above for peptides SEM1(45–67) and SEM1(49–67). From the set of resulting structures, we selected ones that are characterized by the lowest energy. Next, the all-atom molecular dynamics (MD) simulation of these structures was performed using Gromacs (v. 2022) software [41] with NMR distance constraints data for SEM1(45–107) and SEM1(49–107). In the final step of the calculation, MD-trajectories were classified via the clustering analysis tool in GROMACS (gmx cluster) [42] with the corresponding root mean square deviation (RMSD) of Cα atoms cut-off to search preferential ensembles of peptide structures. In the Appendix A, donut plots quantifying the conformational states of SEM1(45–107) and SEM1(49–107) are shown. The diagram represents the fractional occupancies of the four most populated clusters. Each slice represents a distinct state. RMS deviations of Cα-Cα atom-pair distances were used to define the distance between structures.

Figure 10 shows four populated clusters of SEM1(45–107) containing up to 55% of total protein structures from the MD-trajectory of SEM1(45–107), where Figure 10A–D correspond to 21%, 13%, 11% and 9% of total protein structures, respectively. Figure 11 shows four populated clusters of SEM1(49–107) containing up to 75% of total protein structures, wherein Figure 11A, Figure 11B, Figure 11C and Figure 11D correspond to 32%, 16%, 14% and 13% of them, respectively. All ensembles of SEM1(45–107) contain helical fragments in the N-terminus, which is in good agreement with SEM1(45–67) spatial structure data. At the same time, the C-terminus spatial structures of SEM1(45–107) and SEM1(49–107) are similar. Thus, the differences in the N-terminuses of these peptides do not provide significant effects on the spatial structure of the C-terminuses.

## 3. Discussion

The estimation of the translational mobility of peptides via DLS, DOSY and *ζ* -potential measurement shows a stable monomeric form of both SEM1(45–67) and SEM1(49–67). For SEM1(45–67) and SEM1(49–67), we obtained a similar distribution of hydrophobic regions (Appendix A) and electrostatic potential surface distribution of SEM1(45–67) and SEM1(49–67) (Appendix A). Thus, SEM1(45–67) and SEM1(49–67) peptides have a similar behavior in water solution (hydrodynamic radius, *ζ* -potential, hydrophytic surface and electrostatic surface). However, the DLS signal width indicated the less stable conformational state of SEM1(49–67) in comparison with SEM1(45–67).

As it was shown via ThT fluorescence spectroscopy (Chapter 2.1), SEM1(45–107) has a higher amyloid formation rate in comparison to SEM1(49–107) (Figure 2). The various levels of amyloid activity of the studied peptides may be caused by internal differences in their spatial structures. It can be noted that SEM1(45–67) contains helix (E58-K60) and an ordered helix-like structure (S49-Q51) in comparison to SEM1(49–67). In previous articles, the constant secondary structure was observed for individual peptide fragments (PAP(248–261), PAP(262–270) and PAP(274–284)) [43,44], as well as a part of a full-sized PAP(248–286) peptide [45]. Based on the assumption of the secondary structure conservation of individual peptides as well as full-sized protein, the presence of helical fragment SEM1(45–67) in the SEM1(45–107) spatial structure was supposed. Moreover, the MD simulation validates the presence of a helical fragment in the N-terminus of SEM1(45–107) in comparison with SEM1(49–107). The C-terminus (D86-L107) is presented in all amyloidogenic peptides of SEM1. Previously, the C-terminus (D86-L107) spatial structure was published in an article by Sanchugova et al., 2021 [46]. The presence of four extra amino acid residues in the N-terminus of SEM1(45–107) may affect the structure of the C-terminus (D86-L107) with the subsequent difference in the amyloid formation rate of SEM1(45–107) and SEM1(49–107). The peptide structures are necessary to find out this influence. However, the fast aggregation of SEM1(45–107) and SEM1(49–107) makes it impossible to perform NMR structural studies of these peptides. Therefore, we performed MD simulations of SEM1(45–107) and SEM1(49–107) (Figure 10 and Figure 11) to evaluate the influence of a peptide fragment (G45-Y48) on the C-terminus (D86-L107). The analysis of the MD simulations did not show a principal difference between the spatial structure of the C-terminuses (D86-L107) of the SEM1(45–107) and SEM1(49–107) peptides. The helical fragments were present in all configurations of SEM1(45–107), while SEM1(49–107) obtained all configurations characterized by a disordered structure.

The above results raise the question of what breaks the helix of SEM1(49–107), and how is this all connected to the role of the first four residues in the amyloid formation of SEM amyloidogenic peptides. Residue H47 is positively charged and may form electrostatic contact with residue E58 to stabilize the helix motif of SEM1(45–107). Residue 63 is an hydrophobic phenylalanine that may provide hydrophobic contact with the I65 residue side chain in the helix motif that is not observed in SEM1(49–107). Hence, four extra residues stabilize the helix motif of SEM1(45–107), which contributes to forming the stabilized hydrophobic region. Moreover, for other amyloid peptides, it has been previously shown that hydrophobic regions enhance intermolecular cohesion during amyloid formation [47,48]. The higher hydrophobicity of SEM1(45–107) provides the propensity of peptides to form attractive interactions, leading to a fast aggregation rate.

We carried out the CD analysis of SEM1(45–107) fibrils. The recorded CD spectrum (Figure 12) is typical for the β-sheet of amyloids [49]. We proposed that the so-called discordant helix of SEM1(45–107) in the N-terminus may convert to a β-sheet during fibril formation [50]. A more detailed and complete study of this phenomenon is expected in further studies.

## 4. Materials and Methods

### 4.1. Protein Expression and Purification

The expression vectors were obtained by cloning the SEM1(49–107) and SEM1(45–107) peptide fragments of the *H. sapiens* Semenogelin 1 gene fused with 6xHistidine-tagged GB1 partner protein into pET28a plasmid [51]. Histidine-tagged partner protein was linked with semenogelin fragments via a TEV-protease cleavage site for further separation. The expression and purification protocols for the SEM1(49–107) and SEM1(45–107) peptide fragments (SEM1 fragments) are the same and based on the protocol described before with minor modifications [52]. Protein expression was carried out in an *E. coli* BL21 (DE3) pLysS strain (Novagen, Darmstadt, Germany). Cells were grown in LB-rich nutrient medium, supplemented with 50 μg/mL kanamycin and 25 μg/mL chloramphenicol at 37 °C and 180 rpm shaking until the optical density OD_600_ of 0.6–0.8 was reached. The culture was induced to express the SEM1 fragment via the addition of 1 mM isopropyl 1-thio-β-D-galactopyranoside (IPTG) and allowed to grow for 4 h in the same conditions. Then, cells were harvested via centrifugation (5000 rpm, 15 min, 4 °C), frozen and stored at −20 °C.

The frozen cells were thawed, resuspended In buffer 1 (50 mM Tris-HCl, pH 8.8, 0.3 M NaCl) and supplemented with the protease inhibitor cocktail (Roche, Basel, Switzerland) and phenylmethylsulfonyl fluoride (PMSF), followed by sonication on an HD2070 homogenizer (Bandelin, Berlin, Germany) at 4 °C to remove the DNA viscosity. The cell lysate was clarified via 1 h of centrifugation at 100,000× *g* at 4 °C (Ti-45 rotor, Optima XPN centrifuge, Beckman Coulter Inc., Brea, CA, USA) and loaded on NiNTA resin (QIAGEN, Hilden, Germany) equilibrated by buffer 1. SEM1 peptide fragments fused with GB1 and 6xHis-tag were eluted with buffer 2 (50 mM Tris-HCl, pH 8.8; 0.3 M NaCl; 300 mM imidazole). Fractions after elution were pooled and concentrated via Amicon Ultra-4 (10 K) spin-concentrators (Merck, Burlington, MA, USA) to the concentration allowed for loading on the gel filtration column.

Gel filtration was performed using an NGC Discover chromatographic system and Enrich SEC75 column (BioRad, Hercules, CA, USA) in buffer 3 (50 mM Tris-HCl pH 8.5, 0.5 M NaCl) with a 1 mL/min flow rate. Peak fractions were pooled, and the fusion protein was digested via homemade his-tagged TEV-protease [53] at a ratio TEV:GB1-SEM1 fragment equal to 1:100 (*w/w*). Overnight digestion was carried out in the presence of DTT (1 mM), PMSF (1 mM) and EDTA (0.5 mM) at 4 °C [54]. Then, the reaction mix was loaded on NiNTA-resin again to trap GB1 and TEV-protease. The concentration of the SEM1 fragment in the flow-through fraction reached ~2 mM using Amicon Ultra-0.5 (3K) spin-concentrators (Merck, Burlington, MA, USA). The purity of the samples on each purification step was evaluated via polyacrylamide gel electrophoresis under denaturing conditions (SDS-PAGE) in pH 8.3 Tris-glycine buffer [55]. Finally, samples of SEM1(49–107) and SEM1(45–107) peptide fragments with purities of more than 95% were obtained.

The SEM1(45–67) and SEM1(49–67) peptides were synthesized and purified to >95% purity by Almabion (Voronezh, Russia).

### 4.2. ThT Fluorescence

The fibril formation of SEM1(45–107) and SEM1(49–107) was studied using ThT fluorescence intensity measurements. Thioflavin T (ThT) dye fluorescence is regularly used to quantify in vitro amyloid fibril formation. Upon binding to amyloid fibrils, ThT gives a strong fluorescence signal at approximately 482 nm [56]. ThT fluorescence probes were observed at 37 °C on a Thermo Scientific Varioskan LUX multimode microplate reader (Waltham, MA, USA) via FluorEssence (v. 6.1) software in a 96-well microplate. ThT fluorescence assays were prepared by mixing 17 μL of the peptide solution (2mM peptide in 50 mM Tris-HCl pH 8.5, 0.5 M NaCl), 3 μL of 500 μM ThT in 50mM Tris-HCl pH 8.5 and 0.5 M NaCl (total volume of 20 μL) to maintain the fluorescence signal within the linear range of the instrument. The samples were excited at 440 nm, and the fluorescence emission intensity was collected at 482 nm for 90 s and averaged. Fresh peptide solutions of SEM1(45–107) and SEM1(49–107) were prepared for ThT fluorescence measurements. The time interval between the end of synthesis and the registration of fluorescence spectra was less than 10 min. The fluorescence intensity was corrected for lamp intensity fluctuations by dividing the observing fluorescence signal by the lamp intensity. The concentration of SEM1(45–107) and SEM1(49–107) peptides was controlled using NanoDrop One^C^ (Thermo Fisher Scientific, Waltham, MA, USA).

### 4.3. PFG-NMR Spectroscopy

Diffusion experiments were carried out using a 700 MHz NMR spectrometer (AVANCE III-HD, Bruker, Billerica, MA, USA) equipped with a quadruple resonance CryoProbe (^1^H, ^13^C, ^15^N and ^31^P) with a standard z-gradient (a maximum strength of 55.7 G × cm^−1^). Diffusion decays were obtained with the help of the stimulated-echo pulse sequence with water suppression (STEBPGP1S19) containing two field gradient pulses (*g*) with duration *δ*, which are separated by interval Δ [57]. Diffusion decays were fitted by:(2)AgA0=exp−Dγs2g2δ2(∆−δ/3),
where *A*(0) is the spin echo amplitude without gradient pulses,* D_s_* is the self-diffusion coefficient of molecules and *γ* is the proton gyromagnetic ratio. For all experiments, the amplitude of the field gradient pulses (*g*) was varied from 2 to 95% of its maximum under a constant diffusion time (Δ = 100 ms) and gradient pulse duration (*δ* = 3.6 ms). All experiments were performed at 298 K. Data processing and analysis were carried out with Bruker Topspin (v. 3.6) software.

### 4.4. NMR Spectroscopy

NMR experiments of SEM1(49–67) and SEM1(45–67) were carried out with a 700 MHz NMR spectrometer ( AVANCE III-HD, Bruker, Billerica, MA, USA) with a quadruple resonance CryoProbe (^1^H/^19^F, ^13^C, ^15^N and ^31^P) and a 500 MHz NMR spectrometer (AVANCE-II, Bruker, Billerica, MA, USA) with a triple resonance probe TXI (^1^H, ^13^C and ^15^N). For the NMR experiment, peptides were dissolved in water solution (H_2_O + D_2_O/90% + 10%): SEM1(49–67) (0.8 mM, pH 3.5); SEM1(45–67) (0.7 mM, pH 3.0) at 298 K. pH values were chosen to prevent fibril formation [58]. To assign the hydrogen chemical shifts, 2D ^1^H-^1^H TOCSY [59] (TOCSY mixing time d9 = 85ms) and 2D ^1^H-^1^H NOESY [60] (mixing time t_m_ = 350 ms) spectra were obtained. The 2D ^1^H-^15^N HSQC [57] and 2D ^1^H-^13^C HSQC [61], 2D ^1^H-^13^C HMBC [59] and 2D ^1^H-^13^C HSQC-TOCSY [62] spectra were obtained to perform nitrogen and carbon chemical shift assignments, respectively. Scalar coupling constants ^3^J_Hα-HN_ were determined using 2D JRES [63] spectra. The following parameters remained constant in all experiments: the relaxation delay of 1.5s, the number of increments of 2048 × 1024 and the WATERGATE method [57] for water suppression (if needed). The parameters of 2D NMR spectra are shown in the Appendix A.

Data processing was carried out using Bruker Topspin (v. 3.6) software. All spectra were analyzed with the help of the CCPNMR (v. 2.5) program [64].

### 4.5. CD Spectroscopy

CD experiments of SEM1(49–67), SEM1(45–67) and SEM1(45–107) fibrils were performed using a Jasco J-1500 spectrometer (Tokyo, Japan) with a scanning speed of 30 nm/min. For CD experiments, the concentrations of peptides were determined spectrophotometrically using the corresponding extinction coefficients of $E_{214}^{1\%}=28243\;M^{-1}cm^{-1}$, $E_{214}^{1\%}=42598\;M^{-1}cm^{-1}$ and $E_{214}^{1\%}=113084\;M^{-1}cm^{-1}$ for SEM1(49–67), SEM1(45–67) and SEM1(45–107), respectively. The concentration was determined as 4 μM for SEM1(49–67), 5 μM for SEM1(45–67) and 1 μM SEM1(45–107) fibrils, respectively. The peptide extinction coefficient was calculated using the web-service Prot.pi (www.protpi.ch/Calculator/ProteinTool, accessed on 1 December 2022). The CD spectra were recorded in 1 cm path length cells in the 190–240 nm wavelength range at 298 K. The peptide secondary structure was analyzed in the frame of the CDSSTR algorithm (protein set 4 and 7) using DichroWeb server (http://dichroweb.cryst.bbk.ac.uk, accessed on 1 December 2022) [65]. For the CD analysis of SEM1(45–107) fibrils, we mixed a mature SEM1(45–107) fibril solution (Tris buffer, 100 μL) with water (1000 μL) to decrease the effect of Tris on the CD spectrum.

### 4.6. Calculation and Visualization

The spatial structure calculations of SEM1(49–67) and SEM1(45–67) were performed using the XPLOR-NIH (v. 3.6) program [66]. The internuclear distances and ^3^J_Hα-HN_ alpha coupling constants were determined from the analysis of NOESY and 2D JRES, respectively. The dihedral angles were obtained from the TALOS+ (v. 3.8) program [67] using a chemical shift of backbone chain nuclei: H, N, H_α_, C_α_, C. The standard non-linear annealing protocol of the XPLOR-NIH (v. 3.6) software [68] was applied for the calculation of the spatial structure. NMR restraints (internuclear distances and ^3^J_Hα-HN_ alpha coupling constants) were used as input parameters in the XPLOR-NIH calculations. Individual peptide structures were minimized, heated to 1000 K for 6000 steps, cooled in 100 K increments to 50 K, and finally, structures were minimized with 1000 steps of steepest descent followed by 1000 steps of conjugate gradient minimization. Thus, a total of 1000 structures were determined, and they were refined in subsequent calculations using the protein.par force field [68,69]. Furthermore, the water refinement of the calculated structures with additional NMR restraints [70] was performed using XPLOR-NIH. Finally, the 12 (for SEM1(49–67)) and 11 (for SEM1(45–67)) structures with the lowest energy were selected.

The peptide spatial structures were visualized using UCSF Chimera (v. 1.16) [71] and MOLMOL (v. 2K.2) software [72]. Ramachandran’s plot was calculated using the MolProbity website (molprobity.biochem.duke.edu, accessed on 15 December 2022).

### 4.7. DLS Spectroscopy

The diffusion coefficient and hydrodynamic radius in mQ water solutions with concentrations of 0.8 and 1 mg/mL for SEM1(49–67) and SEM1(45–67), respectively, were obtained using DLS Photocor Complex (Moscow, Russia) equipped with a He-Ne laser (λ = 632.8 nm). Cylindrical glass cuvettes with 1 cm radii were used for the DLS measurements. Each correlation function was collected at 90° for 60 s with 20 repeats. We used the CONTIN algorithm to evaluate the corresponding distribution functions [73]. The modified Stokes–Einstein relation was applied to calculate the hydrodynamic radius of molecule *R*:(3)$$R=\frac{k_{B}Tq^{2}}{6\pi \eta }\tau ,$$
where *k* is the Boltzmann constant, *T* is the absolute temperature, *η* is dynamic viscosity of solutions, *τ* are characteristic times,* q* is the scattering vector and *q* = 4*πn*_0_/*λ sin*(*θ*/2) (*λ* = 632.8 nm, *n*_0_ = 1.33 (water),* θ* = 90°, *q*(90°) = 18.5 × 10^4^ cm^−1^).

Zeta potential (*ζ*) was analyzed in aqueous solutions by means of a DLS technique on the Photocor Complex. Experiments were conducted in a 1 cm cell with a carbon electrode for electrokinetic measurements under the same conditions as for the hydrodynamic radius measurements.

### 4.8. MD Simulation

To generate the structural ensembles of the SEM1(45–107) and SEM1(49–107) peptides, we used an integrative approach that incorporates NMR distance constraints into molecular dynamics simulations using a method from Sinelnikova and Spoel’s 2021 article [74]. The all-atom MD simulation of peptides was performed using Gromacs (v. 2022) software [41]. The Charmm36 [75] and TIP3P [76] models were used to simulate protein and water molecules, respectively. The initial conformations of SEM1(45–107) and SEM1(49–107) were prepared using the XPLOR-NIH (v. 3.6) program, where we selected structures of peptides with the lowest energy. These structures were solvated in a rhombic dodecahedron box with an initial volume of 282 nm^3^ containing 9287 water molecules for SEM1(45–107) and 255 nm^3^ containing 8371 water molecules for SEM1(49–107). The neutralization of systems was performed by adding 3 Cl^−^ ions. The obtained systems were minimized using the steepest descent algorithm with a target maximum force of 1000 kJ mol^−1^nm^−1^. In the next step, equilibration was performed in the canonical NVT (constant Number of particles, Volume, and Temperature) ensemble for 100 ps at 300 K using the Berendsen thermostat [77] and for 100 ps at 300 K and 1 bar in the isothermal–isobaric NPT (constant Number of particles, Pressure, and Temperature) ensemble using a Parrinello–Rahman barostat [78]. Furthermore, simulations were carried out for 100 nanoseconds using the same pressure (1 bar) and temperature (300 K) as for the equilibration process. NMR distance constraints were used as input parameters during the MD simulation [74]. After the MD simulation, structural ensembles of SEM1(45–107) and SEM1(49–107) and their MD-trajectory were obtained. In the final stage, to characterize the convergence of structural ensembles, we performed a cluster analysis of the MD-trajectories (Appendix A) via the GROMOS clustering algorithm [79]. GROMOS clustering was performed with 0.5 nm RMSD of the Cα-Cα atom-pair distance cut-off for two structures to be neighbors.

### 4.9. Transmission Electron Microscopy

The presence of fibrils was shown with the help of transmission electron microscopy (TEM) using the Hitachi HT7700 Exalens scanning electron microscope (Tokyo, Japan). Solutions (10 μL) of SEM1(45–107) and SEM1(49–107) peptides in Tris-buffer (10^−3^ M) were placed on a 3 mm formvar/carbon-coated copper grid, and drying was carried out at room temperature. The analysis was carried out at an accelerating voltage of 100 kV in TEM mode.

## 5. Conclusions

In this work, we studied two amyloidogenic peptides of human Semenogelin 1 (residue 45–107, SEM1(45–107) and residue 49–107, SEM1(49–107)) and their N-domains (SEM1(45–67) and SEM1(49–67), respectively). The amyloid fibril formation of SEM1(45–107) and SEM1(49–107) was studied via ThT-fluorescence. It was found that the amyloid formation of SEM1(45–107) starts immediately after purification, whereas the increased ThT fluorescence intensity of SEM1(49–107) solution was observed 30 min after purification, indicating the amyloid formation. The spatial structures of the N-domains of SEM1(45–107) and SEM1(49–107) (SEM1(45–67) and SEM1(49–67)) manifest disordered peptide structures. However, it was found that SEM1(45–67) contains helical fragments in comparison to SEM1(49–67). Previous studies have suggested that helical fragments may rearrange into amyloid β-strands. Thus, the difference in SEM1(45–107) and SEM1(49–107) amyloid-forming behavior may be explained by the presence of the structured helical part of the SEM1(45–107) N-terminus, which contributes to an increased rate of amyloid formation.

## Figures and Tables

**Figure 1 ijms-24-08949-f001:**
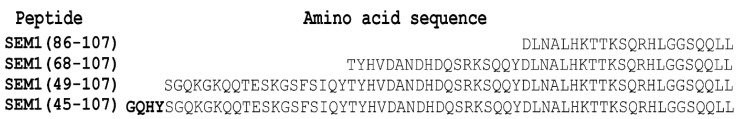
Amino acid sequences of amyloidogenic peptides from Semenogelin 1.

**Figure 2 ijms-24-08949-f002:**
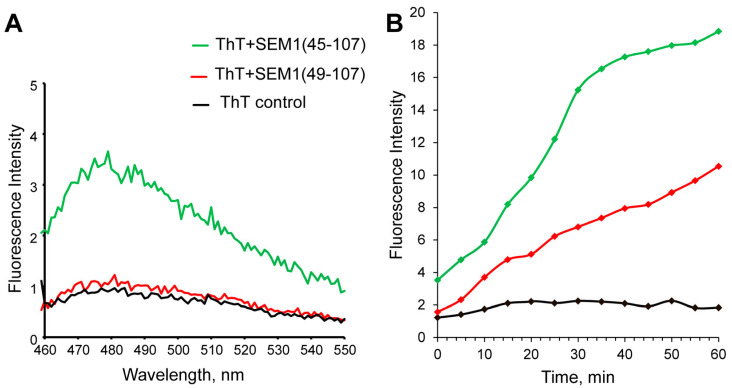
(**A**) Thioflavin T (ThT) fluorescence spectra of SEM1(49–107) (red line), SEM1(45–107) (green line) and control (black line). (**B**) Real-time ThT fluorescence assays of SEM1(49–107) (red line), SEM1(45–107) (green line) and control (black line) solutions.

**Figure 3 ijms-24-08949-f003:**
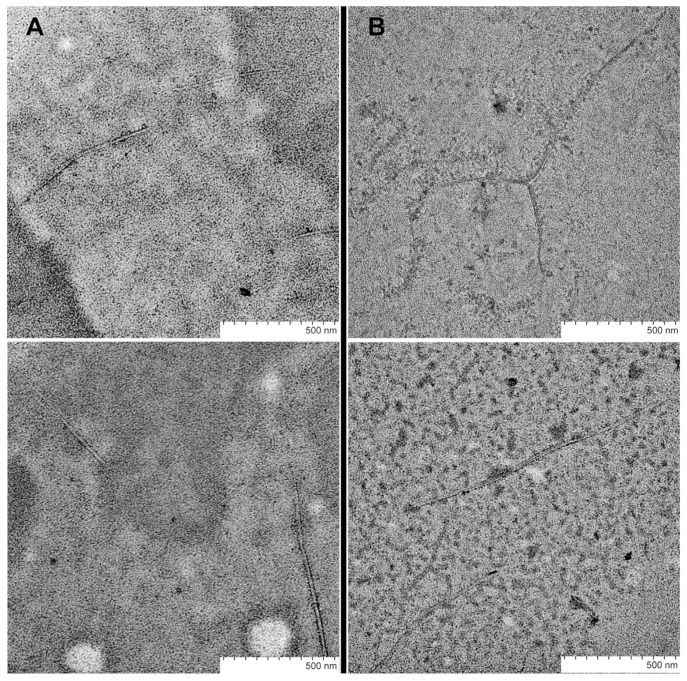
TEM images of SEM1(49–107) (**A**) and SEM1(45–107) (**B**) fibrils.

**Figure 4 ijms-24-08949-f004:**
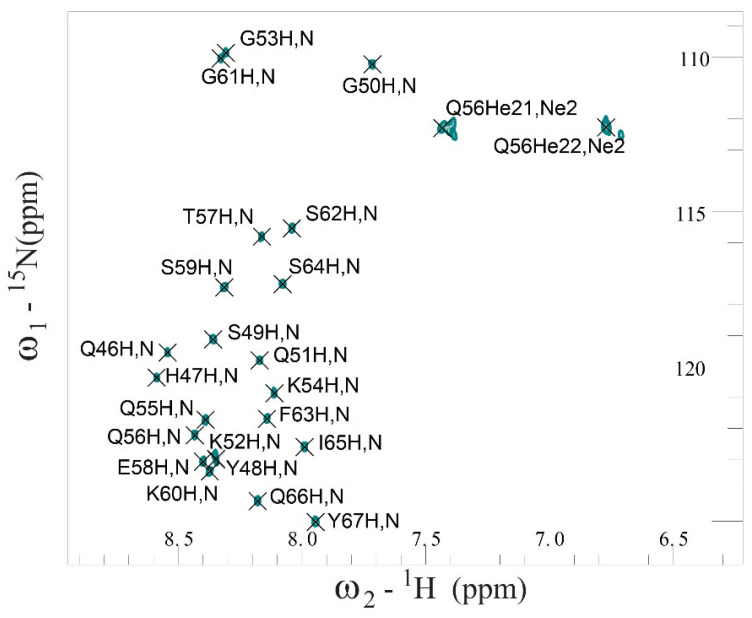
Fragment of 2D ^1^H-^15^N HSQC NMR spectrum of SEM1(45–67) in water solution at 298 K.

**Figure 5 ijms-24-08949-f005:**
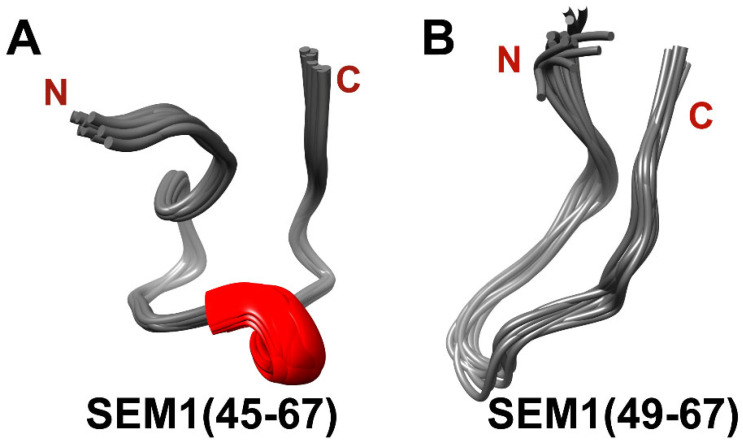
(**A**) Spatial structure of SEM1(45–67) ensemble (11 structures); (**B**) spatial structure of SEM1(49–67) ensemble (12 structures).

**Figure 6 ijms-24-08949-f006:**
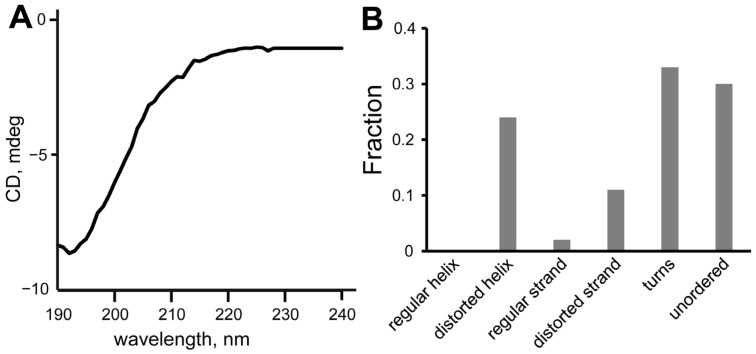
(**A**) CD spectrum of SEM1(45–67), (**B**) component analysis of SEM1(45–67) CD spectrum.

**Figure 7 ijms-24-08949-f007:**
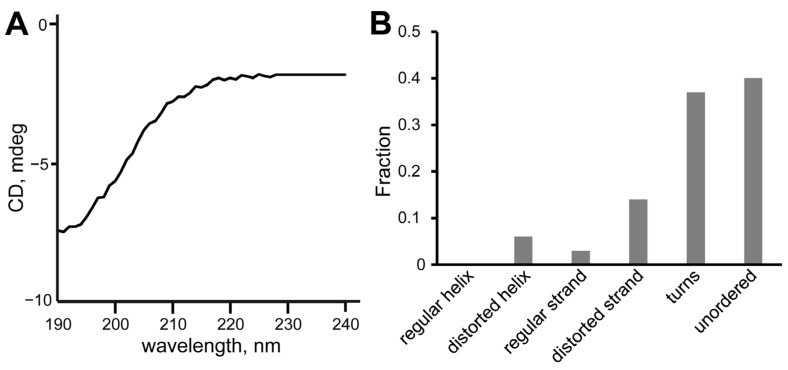
(**A**) CD spectrum of SEM1(49–67); (**B**) component analysis of SEM1(49–67) CD spectrum.

**Figure 8 ijms-24-08949-f008:**
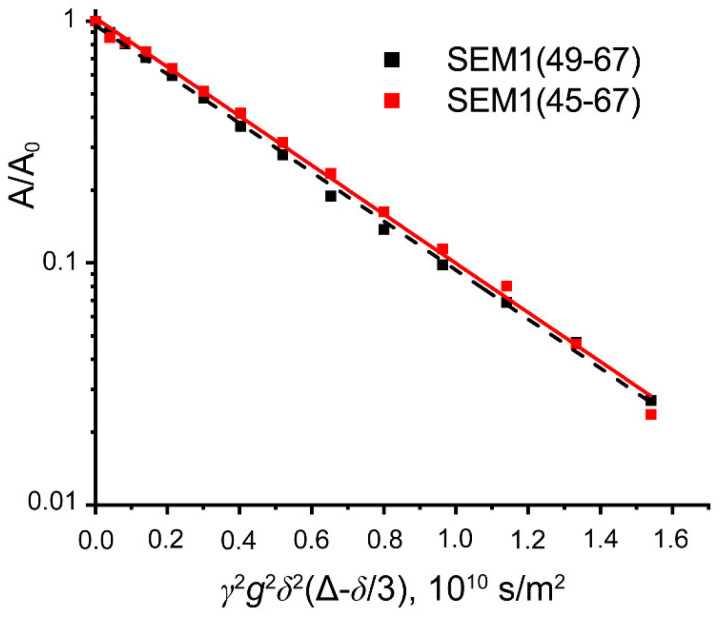
Diffusion decays of signal intensity at 8.02–7.94 ppm for SEM1(45–67) (red) and SEM1(49–67) (black).

**Figure 9 ijms-24-08949-f009:**
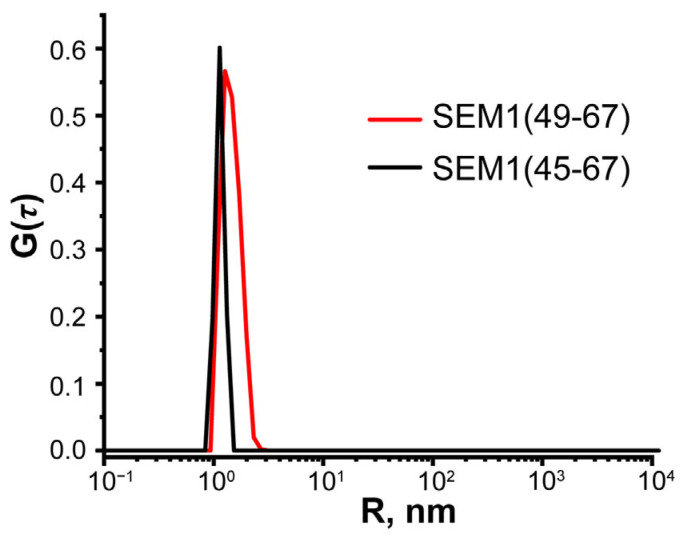
Hydrodynamic radius redistribution for monodisperse water solutions of SEM1(49–67) (red line) and SEM1(45–67) (black line).

**Figure 10 ijms-24-08949-f010:**
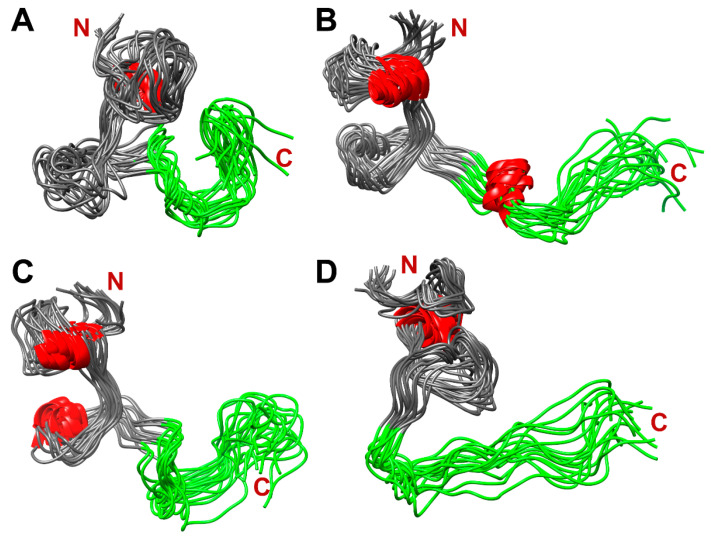
The dominant clusters of SEM1(45–107) holding ~55% of total protein structures: 21% (**A**), 13% (**B**), 11% (**C**) and 9% (**D**). Red labeled fragments are helical fragments, and green labeled fragments are C-terminuses (86–107) of SEM1(45–107).

**Figure 11 ijms-24-08949-f011:**
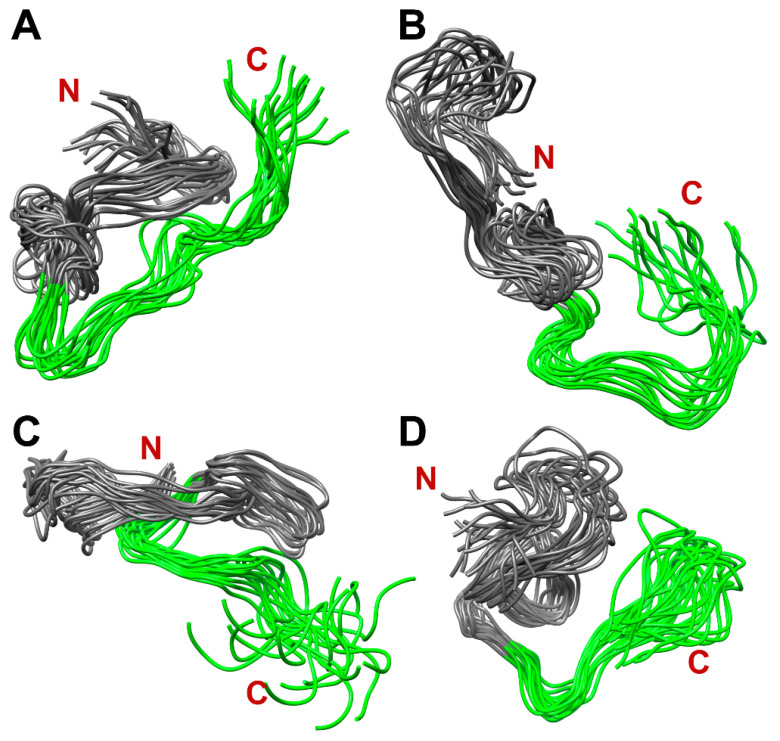
The dominant clusters of SEM1(49–107) holding ~75% of total protein structures: 32% (**A**), 16% (**B**), 14% (**C**) and 13% (**D**). Green labeled fragments are C-terminuses (86–107) of SEM1(49–107).

**Figure 12 ijms-24-08949-f012:**
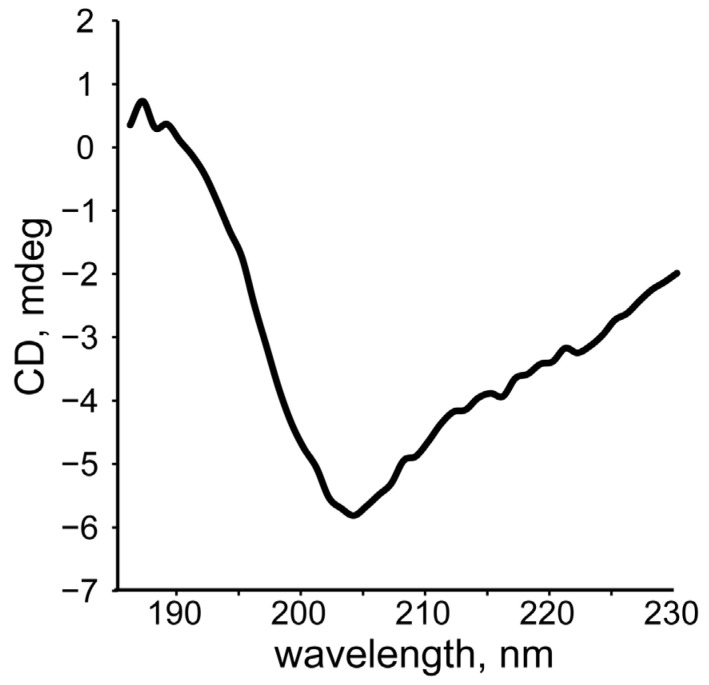
CD spectrum of SEM1(45–107) fibrils.

**Table 1 ijms-24-08949-t001:** Statistical information for SEM1(45–67) structural ensemble.

Distance Constraints	
total	143
intra-residual	88
inter-residual	55
sequential (i–j = 1)	24
medium (i–j = 2–6)	16
long-range (i–j > 6)	15
**^3^J_Hα-HN_ alpha coupling constants (jna)**	
Restraints	22
**Structural Statistics**	
number of NOE violations	0
number of jna restraint violations	0
RMSD for bond deviations (Å)	0.005
RMSD for angle deviations (deg)	0.598
RMSD of all backbone atoms (Å)	
E58-K60	0.13 ± 0.04
S49-Y67	0.39 ± 0.11
Y48-K52	0.14 ± 0.06
**Ramachandran Plot**	
residues in the most favored region (%)	80.1
residues in the additionally allowed region (%)	19.9
residues in the disallowed region (%)	0.0

**Table 2 ijms-24-08949-t002:** Statistical information for SEM1(49–67) structural ensemble.

Distance Constraints	
total	67
intra-residual	54
inter-residual	13
sequential (i–j = 1)	8
medium (i–j = 3)	2
long-range (i–j > 13, 14)	3
**Dihedral Angle Constraints**	
Phi restraints	17
Psi restraints	17
**^3^J_Hα-HN_ alpha coupling constants (jna)**	
Restraints	16
**Structural Statistics**	
number of NOE violations	0
number of dihedral angle restraint violations	0
number of jna restraint violations	0
RMSD for bond deviations (Å)	0.004
RMSD for angle deviations (deg)	0.794
RMSD of all backbone atoms (Å)	
E58-K60	0.24 ± 0.10
S49-Y67	1.12 ± 0.30
Y49-K52	0.69 ± 0.26
T57-S59	0.17 ± 0.08
G50-T57	0.70 ± 0.25
**Ramachandran Plot**	
residues in the most favored region (%)	94.1
residues in the additionally allowed region (%)	5.9
residues in the disallowed region (%)	0.0

## Data Availability

The ^1^H, ^15^N and ^13^C chemical shifts of SEM1(45–67) and SEM1(49–67) have been deposited to BioMagResBank, with BMRB ID 51673 and 51691, respectively. The calculated structures of SEM1(45–67) and SEM1(49–67) have been deposited to the Protein Data Bank, with PDB ID 8BOO and 8BVZ, respectively.

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
