# Peer review of "Extent of N-Terminus Folding of Semenogelin 1 Cleavage Product Determines Tendency to Amyloid Formation"

_ijms, 2023, doi:10.3390/ijms24108949_

Round 1
Reviewer 1 Report
The authors studied the amyloid-formation properties of N-terminal SEM1 in this manuscript. They characterized the aggregation of SEM1(45-107) and SEM1(49-107) using Th-T assay and determined the structures and structural properties of Sem1(45-67) and Sem1(49-67) using NMR spectroscopy and DLS techniques. They concluded that residues 45-48 impact the structure and aggregation properties. Their interesting study provides some new information on the structure of SEM1. However, there are a few serious flaws. This manuscript is recommended for major revision before acceptance for publication.
1. From the present study, the obtained results clearly demonstrated that the C-terminus is the key region for the fibril formation for SEM1. However, the authors do not correlate the impact of residues45-48 on the aggregation of the C-terminus. By the way, A structural study of SEM1(86-107) published in BionanoScience (2021) should be cited in the manuscript.
2. Both SEM1(45-67) and SEM1(49-67) do not form amyloid aggregates. the difference in structure between SEM1(45-67) and SEM1(49-67) is a short helix-like forming at residues S49-Q51. The authors used the conclusion at SEM1(45-67) to account for the fast-aggregation of SEM1(45-107) over SEM1(49-107). I am afraid that the authors might confuse the hypothesis of the helix-enhanced formation of amyloid fibril. This hypothesis only validates under the condition that the proteins or peptides can form an amyloid fibril. It is inappropriate that the authors applied results from a nonamyloid peptide to interpret the mechanism of amyloid formation on an amyloid peptide. Therefore, the authors must provide the same evidence on SEM1(45-107) and Sem1(49-107).
3. Following comments 1 and 2, I suspect that the impact of residues45-48 on amyloid may come from their interaction with the C-terminus and how the effect of helix forming at SEM1(45-107) is therefore discussable.
4. It is not appropriate to interpret Th-T assay just based on the intensity. Many factors can influence the results. A time-dependent Th-T-assay should be included.
5. The calculation of secondary structure using Dicroweb usually gives an RMSD value. The RMSD values should be included in the results.
6. The NOE patterns correlated to the secondary structure should be reported.
7. If possible, the TEM images for amyloid fibril for SEM1 amyloid should be included.
Author Response
Please see the attachment.
Dear Reviewer,
Thank you for constructive and helpful comments and revised the manuscript accordingly. Below we address each of your points.
Our answers are after words “Author reply” in the attachment file, which also contains new text fragments in the manuscript in italic.
Thank you for making their comments and for taking time and effort to help us improve our manuscript.
Sincerely,
Vladimir Klochkov

Reviewer 2 Report
This is a good paper which reports a comprehensive study of two peptides derived from semenogelin. The truncated sequence, SEM1(47-67) was found to be less prone to aggregation than the sequence with four more residues at the N-terminus, SEM1(45-67). Both peptides form hairpin-like structures but the truncated variant has some helix content. The research appears to be sound, the experiments and computations were carried out correclty and the conclusions are justified. State-of-the-art tools were used for experimental data processing and structure determination. The paper is well written and easy to read. I found only a couple of minor grammar/style problems. I have the following minor suggestions:
1. All distance restraints should be listed, perhaps in the Supporting Information. This is important because otherwise the reader cannot assess whether the hairpin-like structures emerge as a result of angular restraints at the turn or as a result of long-range distance restraints. Figures 4 and S1 seem to show only the (1)H-(15)N signals within the same residues (or the respective labels in these Figures are incomplete).
2. The authors state that both peptides are disordered. I think that this statement should be made more precise. There are apparently no long-range hydrogen bonds between the strands but hairpin-like structures are not disordered. Neither of the two peptides appears to form a statistical coil. The "disorder" is rather of local nature; the Ramachandran maps suggest that most of the residues reside in the polyproline II (PII) region.
3. Page 2, lines 68-69: "Herein, we report the discovery of amyloidogenic peptides SEM1(45-107) and SEM1(49-107) is derived from human SEM1 (Figure 1)."
Probably:
"Herein, we report the discovery of amyloidogenic peptides SEM1(45-107) and SEM1(49-107), which is derived from human SEM1 (Figure 1)."
4. Page 4, lines 116-117: "The experimental restrictions for structural calculation"
Probably: The experimental restraints for structural calculation
Author Response

(The authors gave the same response as above.)

Reviewer 3 Report
Osetrina et al. describe structural studies of four fragments of semenogelin protein. Two long ones are found to form amyloid-like aggregates, and two short ones are analyzed in solution by NMR, CD, and DLS.
The authors show that the two short peptides are monomeric under the solution conditions used. But the description of the structural data is confusing and the interpretation is not correct. The CD spectra (very similar to a random coil spectrum) and the low dispersion of the backbone amide NMR signals indicate that the two peptides are not folded, they seem to be sampling many different conformations in fast exchange, a highly dynamic disorder state. There may exist some preferential conformations. But it is not possible to assess their nature from the evidence presented in the manuscript.
Author Response
Please see the attachment.
Dear Reviewer,
Thank you for constructive and helpful comments.
Our answers are after words “Author reply” in the attachment file, which also contains new text fragments in the manuscript in italic.
Thank you for making their comments and for taking time and effort to help us improve our manuscript.
Sincerely,
Vladimir Klochkov

Round 2
Reviewer 1 Report
In the revised manuscript, the authors have made improvements including a time-dependent Th-T assay and MD simulation. However, there are still a few flaws. Therefore, further revision is needed before this manuscript can be accepted for publication.
1. The time-dependent Th-T assay clearly demonstrated that both Sem1(45-107) and Sem1(49-107) could form similar degrees of extended aggregates, but obviously, the aggregation rate of SEM1(45-107) is faster than that of SEM1(49-107). The authors should put more discussion on this point and discuss the role of residues 45-48 on the fast aggregation rate. The time-dependent Th-T assay in supplemental information should replace Figure 3, since it is more informative.
2. Following comment 2, TEM images for both sem1(45-107) and sem1(49-107) at different time points are necessary to further clarify the aggregation states for both peptides.
3. The sections of MD simulation both in methodology and the result are hard to follow. Lots of details are not clear, such as the RMSD profile. An extensive English revision is required.
4. Based on the revised version, the results from the MD simulation raised another question. If the N-terminal conformation of Sem1(45-107) still remains quite stable after the 100ns simulation, then how do the four residues 45-48 accelerate the aggregation rate? From the structures shown in Figure 10, the N-terminus and C-terminus of Sem1(45-107) do not fold to interact with each other. Again, how do the four residues 45-48 enhance the formation of β-strand conformation and aggregation?
5. To verify the results of the MD simulation, the CD spectra for Sem1(45-107) and Sem1(49-107) at different times are needed.
Author Response
Please see the attachment.
Dear Reviewer,
Thank you for the interest in our manuscript. We thank you for constructive and helpful comments and revised the manuscript accordingly. We responded each of the points in Attachment file. We prepared the corrected version of our manuscript. Our answers are after the words “Author reply”. Below we added new text fragments in the manuscript in italic.
Best regards,
Vladimir Klochkov

Reviewer 3 Report
I do not see a substantial improvement in the revised version.
Author Response
Please see the attachment.
Dear Reviewer,
We thank you for your comments. We revised the manuscript. Our answer is after the words “Author reply” in Attachment file. We added new text fragments in the manuscript in italic.
Best regards,
Vladimir Klochkov

Round 3
Reviewer 1 Report
The revised manuscript has been improved a great deal. However, the function and role of residues 45-48 in accelerating amyloid formation are still not clarified. A few points are needed for further revision.
1. The quality of the TEM images for amyloid fibrils is poor.
2. The CD spectra at the aggregation state can help to answer whether the helix formed in SEM1(45-107) converts to β-sheet. If so, the hypothesis of the discordant helix may help to explain the existence of the short helix induced by residue 45-48.
Author Response

(The authors gave the same response as above.)

Round 4
Reviewer 1 Report
The resolution of TEM images has been improved. However, the TEM images shown in Fig. 3 are hard to tell where the amyloid fibrils are. The authors should provide better TEM images for SEM1 fibrils.
Author Response
Dear Reviewer,
We thank you for your comment. Our answer is after the words “Author reply” in Attachment file. We added a revised version of Fig.3.
Best regards,
Vladimir Klochkov
